# The Therapeutic Effects of Dodecaborate Containing Boronophenylalanine for Boron Neutron Capture Therapy in a Rat Brain Tumor Model

**DOI:** 10.3390/biology9120437

**Published:** 2020-12-01

**Authors:** Yusuke Fukuo, Yoshihide Hattori, Shinji Kawabata, Hideki Kashiwagi, Takuya Kanemitsu, Koji Takeuchi, Gen Futamura, Ryo Hiramatsu, Tsubasa Watanabe, Naonori Hu, Takushi Takata, Hiroki Tanaka, Minoru Suzuki, Shin-Ichi Miyatake, Mitsunori Kirihata, Masahiko Wanibuchi

**Affiliations:** 1Department of Neurosurgery, Osaka Medical College, 2-7 Daigaku-machi, Takatsuki-shi, Osaka 569-8686, Japan; neu146@osaka-med.ac.jp (Y.F.); neu149@osaka-med.ac.jp (H.K.); neu142@osaka-med.ac.jp (T.K.); neu133@osaka-med.ac.jp (K.T.); neu130@osaka-med.ac.jp (G.F.); neu106@osaka-med.ac.jp (R.H.); neu-wanibuchi@osaka-med.ac.jp (M.W.); 2Research Center of Boron Neutron Capture Therapy, Research Organization for the 21st Century, Osaka Prefecture University, 1-1 Gakuen-cho, Nakaku, Sakai-shi, Osaka 599-8531, Japan; y0shi_hattori@riast.osakafu-u.ac.jp (Y.H.); kirihata@biochem.osakafu-u.ac.jp (M.K.); 3Institute for Integrated Radiation and Nuclear Science, Kyoto University, 2 Asashiro-Nishi, Kumatori-cho, Sennan-gun, Osaka 590-0494, Japan; watanabe.tsubasa.8x@kyoto-u.ac.jp (T.W.); taku-takata@rri.kyoto-u.ac.jp (T.T.); tanaka.hiroki.3e@kyoto-u.ac.jp (H.T.); suzuki.minoru.3x@kyoto-u.ac.jp (M.S.); 4Kansai BNCT Medical Center, Osaka Medical College, 2-7 Daigaku-machi, Takatsuki-shi, Osaka 569-8686, Japan; rad151@osaka-med.ac.jp (N.H.); neu070@osaka-med.ac.jp (S.-I.M.)

**Keywords:** boron neutron capture therapy (BNCT), malignant glioma, survival prolongation, boronophenylalanine–amide alkyl dodecaborate (BADB)

## Abstract

**Simple Summary:**

We have developed a new boron compound for application in boron neutron capture therapy (BNCT) named boronophenylalanine–amide alkyl dodecaborate (BADB). It is characterized by a larger amount of ^10^B per molecule, linking boronphenylalanine (BPA) and dodecaborate, and we conducted various experiments on its efficacy. Its high accumulation at the cellular level made it a promising novel drug, but it did not sufficiently accumulate in brain tumor tissue when intravenously administered. However, in neutron irradiation experiments, the drug showed remarkably high compound biological effectiveness and significantly prolonged the survival time in rat brain tumor models. We confirmed the antitumor efficacy of BADB in BNCT and its additional efficacy when administered in combination with BPA. Though this drug showed poor results when administered as a single agent, it was superior to BPA alone when administered in combination with BPA, making it a drug that we have been waiting for in our clinical practice.

**Abstract:**

Background: The development of effective boron compounds is a major area of research in the study of boron neutron capture therapy (BNCT). We created a novel boron compound, boronophenylalanine–amide alkyl dodecaborate (BADB), for application in BNCT and focused on elucidating how it affected a rat brain tumor model. Methods: The boron concentration of F98 rat glioma cells following exposure to boronophenylalanine (BPA) (which is currently being utilized clinically) and BADB was evaluated, and the biodistributions in F98 glioma-bearing rats were assessed. In neutron irradiation studies, the in vitro cytotoxicity of each boron compound and the in vivo corresponding therapeutic effect were evaluated in terms of survival time. Results: The survival fractions of the groups irradiated with BPA and BADB were not significantly different. BADB administered for 6 h after the termination of convection-enhanced delivery ensured the highest boron concentration in the tumor (45.8 μg B/g). The median survival time in the BADB in combination with BPA group showed a more significant prolongation of survival than that of the BPA group. Conclusion: BADB is a novel boron compound for BNCT that triggers a prolonged survival effect in patients receiving BNCT.

## 1. Introduction

Boron neutron capture therapy (BNCT) is a type of particle treatment that selectively destroys tumor cells that has been clinically applied to invasive cancers such as high-grade glioma [1,2,3], meningioma [4], melanoma [5], and head and neck cancer [6]. In BNCT, a boron compound that accumulates inside the tumor cells is intravenously injected into the patient. Upon neutron irradiation, the boron-10 (^10^B) atoms are captured by thermal neutrons, producing high linear energy transfer (LET) particles (α particles and ^7^Li recoil nuclei). Since these particles have a short path length (5–9 μm) and their path length approximately corresponds to the size of one tumor cell (10 μm), their disruptive effects are limited to boron-containing cells. Their limited effect within a single cell can selectively destroy tumor cells and spare adjacent cells [7].

At our institution, we have clinically applied BNCT since 2002 as an experimental adjuvant therapy in patients with recurrent or newly diagnosed malignant gliomas. The median survival time (MST) of patients treated with BNCT has been found to be significantly longer than that of historical controls who were treated with surgical removal followed by radiation therapy and chemotherapy at our institute [2,3].

Several types of new boron-containing drugs for BNCT have been proposed to date [8,9,10,11], but clinical experience is limited to the use of boronophenylalanine (BPA), borocaptate sodium (BSH), or both in combination [12]. BPA crosses the blood–brain/–tumor barrier (BBB or BTB) via L-type amino acid transporter 1 (LAT1) by intravenous (i.v.) administration but only contains one ^10^B atom per molecule. In comparison, BSH has twelve ^10^B atoms per molecule but has no specific uptake mechanism and can only be distributed in the brain or brain tumor in cases with a disrupted BBB/BTB.

The development of novel and more efficient boron-carrying agents is an important challenge to overcome in order to make progress with this treatment. The most important requirements for boron compounds in BNCT are (1) a low intrinsic toxicity, (2) a high tumor uptake (>20 μg ^10^B) and a low normal tissue uptake (tumor:normal tissue and tumor:blood boron concentration ratios of >3:1), and (3) rapid clearance from the blood and normal tissues with the maintenance of the boron concentration in the tumor during neutron irradiation [13].

In this study, BNCT was applied using a novel boron compound: boronophenylalanine–amide alkyl dodecaborate (BADB). BADB combines the advantages of having phenylalanine with tumor-directed properties, and it contains a large amount of ^10^B per molecule because it was developed to aim for better boron carrying capacity. We conducted a pharmacokinetic analysis of rat brain tumor models using BADB and examined the effects of neutron irradiation on prolonging survival time.

## 2. Materials and Methods

### 2.1. Boron Compounds

BADB was synthesized and prepared according to the method reported by Hattori et al. (see patent information in Section 6). BADB was synthesized by using phenylalanine—which boasts the same amino acid transporter-directed properties as BPA and dodecaborate—that has a high water-solubility and a low toxicity as a boron carrier. The chemical structure of BADB is shown in Figure 1. It has a molecular weight of 547.82 g/mol and has 13 boron atoms per molecule. BPA (L-isomer) was kindly supplied by Stella Chemifa (Osaka, Japan) and was converted to a fructose complex [14]. All the boron compounds used in this study were ^10^B-enriched.

### 2.2. Cell Culture

The following cell lines were utilized for the cellular uptake of boron experiments: F98 rat glioma, A172 human glioblastoma, C6 rat glioma, B16 mouse melanoma, and SAS human squamous carcinoma. The F98 rat glioma cells produce invasive tumors in the brain of Fisher rats, which have been shown to be refractory to many treatments including radiation therapy. Based on in vivo histology, F98 rat glioma cells have been characterized as progressing to anaplastic or undifferentiated gliomas [15]. In this study, F98 rat glioma cells were kindly provided by Dr. Rolf Barth (Department of Pathology, The Ohio State University, Columbus, OH, USA). A172 human glioblastoma cells were purchased from the Japanese Collection of Research Bioresources (JCRB) Cell Bank, National Institute of Biomedical Innovation (Osaka, Japan). C6 rat glioma, B16 mouse melanoma, and SAS human squamous carcinoma cell lines were kindly provided by Dr. Shin-Ichiro Masunaga (Kyoto University Research Reactor Institute, Osaka, Japan) and were cultured in our laboratory on Dulbecco’s Modified Eagle’s Medium (DMEM) supplemented with 10% fetal bovine serum and penicillin, streptomycin, amphotericin B at 37 °C in an atmosphere of 5% CO_2_. All media materials were purchased from Gibco Invitrogen (Grand Island, NY, USA).

### 2.3. The F98 Rat Glioma Model

All experiments using animals were performed according to the guide for the care and use of laboratory animals approved by the Animal Use Review Board and Ethical Committee of Osaka Medical College (permit no. 30036) and the Institute for Integrated Radiation and Nuclear Science, Kyoto University (KURNS; Kumatori, Osaka, Japan) (permit no. 2018-9). The animals used for the study were eight-week-old male Fischer rats weighing approximately 200–250 g (Japan SLC; Shizuoka, Japan). Each rat was anesthetized by an intraperitoneal injection of three different anesthetics mixed together: medetomidine (ZENOAQ, Fukushima, Japan) (0.4 mg/kg), midazolam (SANDOZ, Yamagata, Japan) (2.0 mg/kg), and butorphanol (Meiji Seika, Tokyo, Japan) (5.0 mg/kg). The animal’s head was fixed using a stereotactic frame (Model 900; David Kopf Instruments, Tujunga, CA, USA). The F98 tumor cell implantation into the rat brain was conducted using a surgical procedure that was previously adopted by our research group to confirm the efficacy of a novel boron compound [16,17]. The F98 cells—diluted in a 10 μL solution of DMEM containing 1.4% agarose (Wako Pure Chemical Industries, Osaka, Japan) at a concentration of either 10^3^ cells for the therapeutic experiments or 10^5^ cells for the biodistribution experiments—were injected at a rate of 20 μL/min by an automatic infuser pump.

### 2.4. Cellular Uptake of Boron in F98 Glioma Cells

In this study, BADB exposure was used to determine whether exposure to each boron compound was effective for the uptake of boron into the cells. The following cell lines were used for the in vitro boron-uptake studies: F98, A172, C6, B16, and SAS. These cells were used for the cellular boron-uptake initially studied at Osaka Prefecture University.

From the results, boron uptake was further observed in other cell lines, including F98 glioma cells, after exposure to BADB. First, 10^5^ F98 glioma cells were seeded into a 100-mm dish (Becton, Dickinson, and Company, Franklin Lakes, NJ, USA) with the culture medium described above at 37 °C in a 5% CO_2_ atmosphere. After incubation for three days at 37 °C, the medium was replaced with a culture medium (described above) containing 5, 10, or 20 μg B/mL of BPA or BADB, and the cells were incubated for an additional 2.5, 6 and 24 h at 37 °C.

Separately, in another experiment used to measure the cellular clearance of boron, a cell culture medium containing 20 μg B/mL of BPA or BADB was incubated for 6 h. Then, the medium was replaced with a boron-free medium and incubated for a further one or four hours.

The medium containing the boron compounds was then removed, and the dish was washed twice with 4 °C phosphate-buffered saline (PBS). Then, the cells were detached with trypsin-ethylenediamine tetraacetic acid solution and collected. PBS was then added, and the cells were centrifuged twice (200× *g* for five minutes), counted, and sedimented. The cells were then lysed in a 1 N nitric acid solution (Wako Pure Chemical Industries, Osaka, Japan) overnight. The cellular boron concentrations were measured using inductively coupled plasma atomic emission spectroscopy (ICP-AES: Hitachi, Tokyo, Japan).

### 2.5. The Cellular Distribution of Boron Compounds in F98 Glioma Cells

To evaluate the cellular distribution of boron compounds, 35-mm glass-bottomed dishes (Iwaki, Tokyo, Japan) were seeded with F98 glioma cells (0.5 × 10^5^ cells suspended in 3 mL of DMEM) and allowed to settle for 24 h at 37 °C. The medium was replaced with an equivalent medium containing a boron concentration or with no boron as a control group (the final concentration of BADB was 0.1 mM and that of L-BPA was 1.0 mM in each case), and the cells were cultured for 24 h at 37 °C. DRAQ5^TM^ (BioStatus Ltd., Leicestershire, UK) (with a final concentration of 5 µM) was added to the medium, and the cells were cultured for 30 min at 37 °C. After being washed with DMEM, the medium was replaced with 1 mL of DMEM that contained a fluorescent probe for boronic acid (DAHMI) [18] (with a final concentration of 1.0 mM) before being allowed to settle for 20 min at 37 °C. The distribution of the boron compounds in the cells was analyzed under excitation at 405 nm using an LMS 700 confocal scanning laser microscope (Carl Zeiss, Oberkochen, Germany) equipped with a 20 × objective lens.

### 2.6. Neutron Irradiation in the In Vitro Study

The cytotoxicity due to the capture reaction of each boron compound upon neutron irradiation was evaluated via a colony-forming assay. F98 glioma cells were divided into five groups and maintained as monolayers in 150-cm^2^ tissue culture flasks with 20 mL of medium as follows:
▪Group 1: boron-free medium (irradiated group).▪Group 2: medium containing 10 µg B/mL of BPA for 2.5 h (BPA for 2.5 h group).▪Group 3: medium containing 10 µg B/mL of BADB for 2.5 h (BADB for 2.5 h group).▪Group 4: medium containing 10 µg B/mL of BADB for 24 h (BADB for 24 h group).▪Group 5: medium containing 10 µg B/mL of BPA for 2.5 h after incubating in a medium containing 10 µg B/mL of BADB for 24 h (BADB for 24 h and BPA for 2.5 h group).

Cells were grown in a humidified 5% CO_2_ environment at 37 °C. Neutron irradiation was performed at a neutron flux of 7.0 × 10^8^ neutrons/cm^2^/s for 10, 30, and 60 min at the KURNS. The number of cells in each sample was counted after neutron irradiation and diluted to ensure that the appropriate number of cells was seeded in the 60-mm dish (Becton, Dickinson, and Company Franklin Lakes, NJ, USA) (three replicates each in a dish). The cells were incubated at 37 °C in a 5% CO_2_ atmosphere for seven days, fixed with 10% formalin, and stained with trypan blue. The survival fraction (SF) was calculated by counting the number of colonies consisting of more than 50 cells and dividing that by the number of colonies in the control group. In the present study, a linear–quadratic (LQ) model was employed to fit the SF of the colony-forming assay from X-ray irradiation with F98 glioma cells. The relative biological effectiveness (RBE) of the beam and the compound biological effectiveness (CBE) for each boron compound were determined with SF = 0.1. The RBE value of the neutron beam was calculated by comparing the neutron beam dose and the γ-ray dose in the absence of boron [19].

### 2.7. Biodistribution of Boron in F98 Rat Brain Tumor Model

Each boron compound was administered into the F98 glioma-bearing rat brain tumor models, and the boron uptake into organs and the tumor was evaluated. The F98 glioma-bearing rat brain tumor models were anesthetized 10 days after tumor implantation using the above-mentioned methods and were treated with various boron compounds. Six groups of animals were involved: three groups were i.v. injected with BPA at a concentration of 12 mg B/kg of the rat’s body weight (b.w.), while the other three groups were given BADB by convection-enhanced delivery (CED) (1.2 mg B/kg b.w.) by means of an Alzet osmotic pump (model #2001D; DURECT Corporation, Cupertino, CA, USA). CED is a drug-delivery method for the direct infusion of medication into a tumor-invaded brain by generating a pressure gradient at the tip of an infusion catheter. Using this method, invaded tumor cells could be targeted even in the surrounding brain with an intact BBB [17,20]. Though the intravenous administration of BADB at 2 h after in our pilot study prior to the start of this study resulted in a low uptake of boron into the tumors of the F98 rat brain tumor model and the ratios of boron concentration to normal brain (T/Br ratio) and to blood (T/Bl ratio) were insufficient as low as 1.9 and 0.8, respectively, the uptake in the cells was sufficient enough that we used CED as the route of administration for BADB. Rats were sacrificed at 2 and 6 h after i.v. administration and at 2, 6, and 24 h following the termination of BADB administered by CED. The rats were euthanized, and the tumor, normal brain, blood, heart, lung, liver, spleen, kidney, skin, and muscle were collected and weighed. The amount of boron in each organ was quantified by ICP-AES.

### 2.8. Survival Analysis of In Vivo Neutron Irradiation Study

The therapeutic effect on in vivo neutron irradiation after the administration of each boron compound was evaluated by comparing survival time. BNCT was performed for 14 days following the stereotactic implantation of 10^3^ F98 glioma cells. Forty glioma-bearing rats were randomly divided into the following six groups consisting of 5–8 animals each, with an equal distribution of body weights:▪Group 1: untreated control group.▪Group 2: neutron irradiation only.▪Group 3: BADB-only group.▪Group 4: neutron irradiation following BPA administered by i.v. injection (BPA BNCT group).▪Group 5: neutron irradiation following BADB administered by CED (BADB BNCT group).▪Group 6: neutron irradiation following the combination of BADB administered by CED and BPA administered by i.v. injection (combination of BADB and BPA BNCT group).

After the animals were anesthetized using the above-mentioned mixed anesthetics, their bodies (except for the head) were shielded with ^6^LiF ceramic tiles to shield the thermal neutrons in order to reduce whole-body exposure. At 2 h after the termination of i.v. or 6 h after the termination of CED, the rats were irradiated at a reactor power of 1 MW for one hour. After neutron irradiation, the experimental animal group was left at the KURNS for observation. The therapeutic effects were evaluated in terms of the survival time of all rats. The percentage of increased lifespan (%ILS) was also determined based on the MST, as in our previous research [21].

### 2.9. The Estimation of Physical Dose and Biologically Photon-Equivalent Dose

The radiation dose analysis of BNCT experiments involved the estimated physical dose and biological photon-equivalent dose delivered to the rats after neutron irradiation. The BNCT dose was equal to the sum of the four individual physical dose components (boron, nitrogen, hydrogen, and gamma ray), which were calculated using a method that was previously used by our research group [16,17]. The differences in the neutron spectrum between the surface and the tumor site were simulated and corrected for using a general Monte Carlo N-particle transport code [22]. The simulations were performed while assuming the dimensions of the rat head as described by Futamura et al. [16]. Meanwhile, the biologically photon-equivalent total dose in BNCT (photon-equivalent dose; Gy-Eq) was estimated using a method that was also used previously by our research group [16,17]. In the in vitro neutron irradiation study, the RBE values for nitrogen (RBE_N_) and hydrogen (RBE_H_) for the mixed neutron beam were the values of the physical dose of the beam at SF = 0.1 for each compound exposure condition. The physical dose and photon-equivalent dose for each boron compound was evaluated. The CBE in each boron compound was calculated using the calculated RBE, the physical dose, and photon-equivalent dose. In the in vivo neutron irradiation study, the beam components (e.g., RBE) in the rat brain tumor model could not be calculated differently from that in the in vitro neutron irradiation study, so we used previously reported values [23]. The CBE for each boron compound was calculated using the values calculated in this in vitro study.

### 2.10. Statistical Analysis

Statistical analyses were performed using the JMP^®^ Pro version 14.2.0. software program (SAS, Cary, NC, USA). Comparisons of the cellular uptake of boron in the context of BPA or BADB were conducted using the Student’s *t*-test. The overall survival (OS) from the date of F98 tumor cell implantation was calculated using the Kaplan–Meier method, and significant differences in OS were determined using the log-rank test. Finally, *p*-values of less than 0.05 were considered to be statistically significant.

## 3. Results

### 3.1. Cellular Uptake of Boron in F98 Glioma Cells

The boron concentrations of the five different cells lines are shown in Figure A1. The cellular boron concentrations obtained using BPA and BADB, incubated with 5 and 20 µg B/mL are shown in Figure 2A–C, respectively. In 5 µg B/mL of each boron compound, BPA showed higher concentrations (31.5 ± 1.9, 30.9 ± 1.5, and 38.1 ± 3.8 μg B/10^9^ cells) compared with BADB (14.3 ± 0.5, 12.0 ± 0.8, and 13.3 ± 1.1 μg B/10^9^ cells; *p* = 0.0004) at all incubation times. In 10 µg B/mL of each boron compound, BPA and BADB showed no significant differences in concentration at all incubation times (43.6 ± 4.2, 48.0 ± 5.4, and 52.8 ± 2.2 μg B/10^9^ cells for BPA vs. 51.6 ± 5.0, 46.4 ± 1.1, and 53.4 ± 4.7 μg B/10^9^ cells for BADB). In 20 µg B/mL of each boron compound, BPA and BADB showed no significant differences in concentration at all incubation times. (60.4 ± 14.5, 61.6 ± 13.8, and 94.2 ± 3.9 μg B/10^9^ cells for BPA vs. 83.4 ± 17.0, 72.2 ± 7.9, and 95.7 ± 11.3 μg B/10^9^ cells for BADB) The retention rate of boron for BPA was 38.0% and 55.3% for BADB after one hour of additional incubation, when the medium was changed to boron-free. After four hours of additional incubation, the retention rate of boron for BPA was 33.1% and 34.5% for BADB (Figure 2D).

### 3.2. The Cellular Distribution of Boron Compounds in F98 Glioma Cells

In an image analysis of the fluorescence microscopy findings, using DAHMI as the boron sensor and with the nucleus labeled with DRAQ5^TM^, the distributions of BADB were found to be different from those of BPA. BPA was distributed homogeneously over the cell nucleus and cytoplasm (Figure 3A–C). In contrast, BADB was localized in the cytoplasm with little distribution in the nucleus (Figure 3D–F).

### 3.3. Neutron Irradiation in the In Vitro Study (Colony-Forming Assay)

The results from the in vitro neutron irradiation study are shown in Figure 4. Notably, the SF of F98 glioma cells decreased as the radiation dose increased. For a given dose, the SF of the irradiated control group was higher than those irradiated with the boron compound group. The SF of BPA for 2.5 h group and BADB for 2.5 h group were not significantly different (10 min: 0.502 vs. 0.549; 30 min: 0.140 vs. 0.127; 60 min: 0.057 vs. 0.085). Moreover, for the SF of the BADB for 24 h group, the BADB for 24 h and BPA for 2.5 h group, the differences were even smaller (10 min: 0.251 vs. 0.261; 30 min: 0.023 vs. 0.032; and 60 min: 0.006 vs. 0.006 (BADB for 24 h vs. BADB for 24 h and BPA for 2.5 h), respectively). The LQ model obtained by X-ray irradiation of the F98 glioma cells was used as a method for estimating the doses to calculate the CBE factors for the BPA and BADB groups from the present colony-forming assay. The physical doses required to achieve comparable biological effects (estimated from the survival fractions obtained by the colony-forming assay after neutron irradiation with BPA and BADB, as well as the corresponding baseline LQ model) of the BPA for 2.5 h, BADB for 2.5 h, and BADB for 24 h groups were 3.42, 3.76, and 1.86 Gy, respectively. Meanwhile, the CBE values of the BPA for 2.5 h, BADB for 2.5 h, and BADB for 24 h groups (determined by the calculated RBE) were 2.02, 1.75, and 4.66, respectively.

### 3.4. Uptake of Boron in F98 Glioma-Bearing Rats

The boron concentrations of the tumor, normal brain, and blood in F98 glioma-bearing rats after both the i.v. BPA and BADB administration of 12 mg B/kg and CED BADB administration of 1.2 mg B/kg are summarized in Table 1. The boron concentrations in the tumor at 2 and 6 h after i.v. BPA administration were 17.8 ± 1.4 and 13.4 ± 2.1 µg B/g, respectively. Meanwhile, those at 2, 6, and 24 h after the termination of CED were 32.8 ± 16.7, 45.8 ± 16.5, and 25.1 ± 26.6 µg B/g, respectively. The tumor/normal brain (contralateral brain) ratios at 2, 6, and 24 h after BADB termination were 32.8, 35.2, and 125.5, respectively, while the tumor/blood ratios at 2, 6, and 24 h after BADB termination were 65.6, 114.5, and 125.5, respectively. The boron concentrations with BPA and BADB in other organs were low enough so as to be nearly undetectable (<0.5 µg B/g) (data not shown).

### 3.5. Survival Analysis of the In Vivo Neutron Irradiation Study

The physical and photon-equivalent doses delivered to rat brain tumors during the in vivo neutron irradiation experiments were calculated using the ratio of biological effectiveness of each boron compound obtained in the in vitro neutron irradiation study as a reference and the mean boron concentrations in tumor tissue obtained in the aforementioned biodistribution studies. The RBE (RBE_N_ and RBE_H_) for photon-equivalent dose calculations was 3.0, which was reported previously [23]. The physical dose and photon-equivalent dose of BADB were based on a 25% boron contribution from CED administration. The calculated physical dose was 1.0 Gy for the irradiated group, 3.4 Gy for the BPA i.v. group, and 2.5 Gy for the BADB group. The results of the photon-equivalent dose were 1.7 Gy-Eq for the irradiated group, 6.7 Gy-Eq for the BPA i.v. group, and 9.1 Gy-Eq for the BADB group (Table 2).

Figure 5 indicates the estimated OS times for the F98 rat brain tumor models treated with BNCT. The MSTs of the control groups (untreated, irradiation-only, and BADB-only groups) and the neutron irradiation groups with boron compounds (BPA BNCT, BADB BNCT, and the combination of BADB and BPA BNCT groups) were 26.5 days (25–28 days), 28 days (26–30 days), 28 days (27–29 days), 34 days (33–36 days), 31 days (29–35 days), and 38 days (36–40 days), respectively. The neutron irradiation groups with the boron compound had significantly longer survival times than the untreated control group (vs. BPA i.v., *p* = 0.0006; vs. BADB, *p* < 0.0001; and vs. the combined group, *p* = 0.0002). Among the neutron irradiation groups with the boron compound, the BADB BNCT group showed no significant increase in the survival time than that of the BPA BNCT group (*p* = 0.039). Meanwhile, the combination of BADB and BPA BNCT group boasted longer survival times than the BPA BNCT group (*p* = 0.001). In addition, the combined group showed the highest %ILS value among all groups (43.4%) (Table 3).

## 4. Discussion

BNCT has previously been clinically studied and is now expected to be the next generation of minimally invasive treatment with less burden on patients. It is being aimed at approval as a cancer therapy. To further enhance the efficacy of BNCT, we developed BADB as a novel boron delivery agent and performed a basic study of its efficacy. BADB is a boron carrier that not only has amino acid transporter-directed properties but also contains more ^10^B atoms per molecule than BPA, which has been used for many years in clinical studies using nuclear reactors.

From the in vitro cellular uptake of boron experiment, BADB showed significantly lower concentrations of boron than BPA at 5 µg B/mL, yet the concentrations were mostly equivalent at 10 and 20 µg B/mL. These compounds showed an increase in boron accumulation in cells in a concentration-dependent manner. These results suggested that BADB is more beneficial in higher concentrations. When each molecular weight was considered, BADB was superior to BPA in its ability to deliver boron to cells per molecule. The retention rate study (Figure 2D) showed that BPA leaked out at a time point earlier than BADB and the retention of BADB was better than that of BPA. LAT1 drains intracellular amino acids when it uptakes extracellular BPA, but it also drains intracellular BPA when it uptakes extracellular amino acids [25]. BADB showed a higher retention early after the end of exposure, suggesting that BADB is a suitable boron compound in BNCT.

The cellular distribution study (Figure 3) showed that the distribution of BADB was localized in the cytoplasm and was different from that of BPA. BPA was observed to be widely distributed in the cytoplasm and the cell nuclei, as previously reported [26,27]. The selective uptake of the boron component in BNCT for the treatment of brain tumors has been previously reported in relation to transporter-targeted nano-drug delivery and internalization through transporter-mediated endocytosis [28]. Additionally, BPA is never solely dependent on LAT1 for uptake, and other routes exist for the binding of BPA to cells. All of them involve amino acid transporters, but some routes are concentration-dependent and their details have not been clarified [29,30]. A polymer modified with L-phenylalanine could facilitate the interaction of LAT1, resulting in LAT1 mediated endocytosis [31]. The improvement of retention is a feature in the uptake of boron using endocytosis, with the same mechanism considered to be present in the improvement of the retention of BADB.

In the neutron irradiation in vitro study, the SF values obtained from the colony-forming assay in the neutron irradiation with the BPA and BADB for 2.5 h group were not significantly different. Based on the results of the in vitro cellular uptake and in vitro neutron irradiation experiments, the uptake and cell-killing effects of BPA and BADB were not significantly different for the same exposure time (2.5 h) and boron concentration (10 μg B/mL) of BPA and BADB. The results of the cellular uptake experiments showed that, although the boron concentrations in the 2.5 and 24 h BADB exposure groups were not significantly different, the neutron irradiation experiment revealed an increase in the cell-killing effect with a longer exposure. The addition of BPA for 2.5 h to BADB for 24 h of exposure did not show the enhancement of the cell-killing effect. The results suggest that BADB—which was distributed on the cell membrane at 2.5 h of exposure—underwent endocytosis over a longer period of time (24 h of exposure), that boron was distributed inside the cells, and that the cell-killing effect of BADB was considered to increase with the changing cellular boron distribution. There was no significant contribution in BPA to the cellular-level cell-killing effect of the combination of 24 h of BADB exposure and 2.5 h of BPA exposure. These results suggest that additional exposure to BPA does not alter the cellular-level cell-killing effect of neutron irradiation.

According to the results of the survival analysis in the in vivo neutron irradiation study using the F98 rat brain tumor model, if the tumor boron concentrations from BADB administered by CED directly contributed to the cell-killing effect, then the MST would be prolonged relative to that affiliated with BPA-alone administration. However, the actual result differed from what was expected. Of the 31.3 Gy-Eq photon-equivalent dose calculated based on the boron concentration from BADB administered by CED, the dose that contributed to the antitumor effect was probably less than the photon-equivalent dose of 6.7 Gy-Eq in the BPA i.v. administration group. Considering the absorbed dose delivered into the tumor in our previous research, the fraction of boron responsible in the cell-killing effect of the boron-containing compounds administered by CED was calculated to be low, with a contributing proportion of 26.8% [16,17]. Therefore, it should be noted that, when boron compounds are administered by CED, the boron concentration in the entire tumor tissue is apparently higher and cannot be directly translated into a therapeutic dose with BNCT [17]. In this study, the CED used to administer BADB into the tumor may have shown variable exposure times and concentrations. The photon-equivalent dose to tumor tissue was estimated as 9.1 Gy-Eq when the contributed boron dose of BADB in vivo BNCT was assumed to be 25% from the biodistribution study. The photon-equivalent dose of tumor tissue at a lower estimate was 3.5 Gy-Eq using the boron concentration minus one SD of 29.3 µg B/g for the BADB obtained from the in vivo biodistribution study and the CBE value of 1.75 obtained from the in vitro neutron irradiation study. Thus, it is conceivable that biological effectiveness would vary in the tumor tissue when BADB administered by CED and tumor cells would be expected to have varying contributing doses ranging from 3.5 to 9.1 Gy-Eq. These results suggest a difference in the antitumor effect of BADB because the photon-equivalent dose of whole tumor tissue was lower than the photon-equivalent dose of 6.7 Gy-Eq under BPA i.v. administration.

In the in vitro neutron irradiation study, 24 h of exposure to BADB followed by an additional 2.5 h of exposure to BPA did not contribute to the cell-killing effect, while in vivo neutron irradiation ensured a prolonged survival time, thus indicating a contribution to the antitumor effect. CED allows for the direct infusion of drugs into a tumor or brain parenchyma using pressure-driven bulk flow without having to pass the drug through the BBB, thus providing superior pharmacokinetic advantages over i.v. administration [13]. Though findings of higher boron concentrations in the tumor are expected by using CED, it is necessary to evaluate whether sufficient concentrations of boron can be delivered to achieve a therapeutic effect on boron neutron capture reaction. The macroscopical distribution of boron administered by CED may be different from that of i.v. administration, which may have affected the additional antitumor effect in this study. Therefore, the contribution of BPA to tumor cells that had been insufficiently exposed to the drug when BADB was administered by CED was considered to have increased the antitumor effect of all tumor tissues. A different dosing protocol from the present study with CED (i.e., further prolonged exposure and higher compound boron concentrations) may also improve cellular distribution and further increase the antitumor effect in single agents. In summary, we suggest that BADB may be useful as a novel compound for BNCT because of its high affinity and retention in tumor cells, and its availability in combination with BPA.

To date, various new boron compounds ranging from small molecules to polymers have been reported. For example, there are liposomes containing boron compounds [32,33,34,35,36], core-polymerized and boron-conjugated micelles [37,38], a BSH-fused cell membrane-penetrating peptide [39], and protein-bound boron compounds that rely on the tumor accumulation effect [40] of serum albumin [41], all of which have shown antitumor effects in in vivo experiments. To perform effective BNCT, it is required to develop a boron compound that has less cytotoxicity itself but which can be selectively accumulated at high concentration levels only in tumor cells. The new boron compound, BADB, has the part of the BPA structure that has already been shown to be low in toxicity and high in tumor accumulation and that is characterized by a high boron content and a high retention in the tumor cells.

There are some uncertainties regarding the low uptake of BADB into tumor tissues after i.v. BADB administration, such as its larger molecular weight than that of BPA, while the presence of the BBB or BTB limits the uptake of BADB into tumor tissues. The issue with the use of CED is that though it is currently used in clinical trials including pontine gliomas and spinal cord tumors, cases are few and uncommon. As such, the limitation is that it is unclear how large doses of the drug can be delivered and how catheters can be placed after surgery for brain tumors. Because one of the important advantages of BNCT is its low invasiveness, the safety of CED administration should be investigated, and the correct CBE values should also be determined for normal tissue in BNCT with BADB. We are now considering further studies to introduce the use of BADB into clinical practice. The adoption of this novel boron compound may significantly contribute to the future progress of BNCT.

## 5. Conclusions

BADB is a novel boron compound that shows antitumor effects in BNCT. The combination of BADB administered by CED and BPA administered by i.v. injection had longer survival times relative to a single agent BPA i.v. group. This study suggests that BADB has an additional effect on the prolonged survival by BNCT for glioma, and it has a significant potential to contribute to the future development of this field.

## 6. Patents

Y. Hattori, K. Uehara, M. Kirihata, Preparation of boron containing compounds for boron neutron capture therapy (BNCT) in cancer therapy, PCT Int. Appl. 2018, WO 2018021138 A1 20180201

## Figures and Tables

**Figure 1 biology-09-00437-f001:**
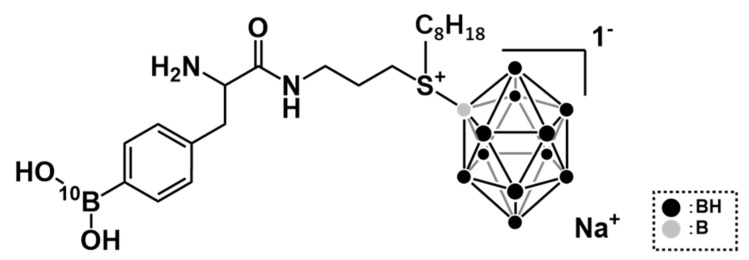
The chemical structure of boronophenylalanine–amide alkyl dodecaborate (BADB).

**Figure 2 biology-09-00437-f002:**
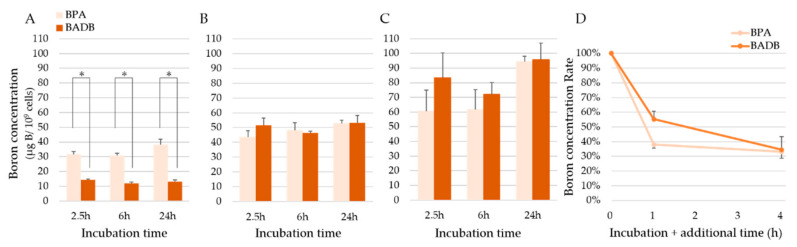
Cellular uptake of boron in F98 glioma cells: (**A**) To discern the cellular uptake of boronophenylalanine (BPA) and BADB, F98 cells were incubated with 5 µg B/mL of BPA and BADB for 2.5, 6, and 24 h. * *p* < 0.005 by Student’s *t*-test. (**B**) To discern the cellular uptake of BPA and BADB, F98 cells were incubated with 10 µg B/mL of BPA and BADB for 2.5, 6, and 24 h. (**C**) To discern the cellular uptake of BPA and BADB, F98 cells were incubated with 20 µg B/mL of BPA and BADB for 2.5, 6, and 24 h. (**D**) To discern the retention of the boron concentration in BPA and BADB, F98 cells were incubated with 20 µg B/mL of BPA and BADB for 6 h with additional incubation with a boron-free medium for zero, one, or four hours The bar in each result indicates the standard deviation.

**Figure 3 biology-09-00437-f003:**
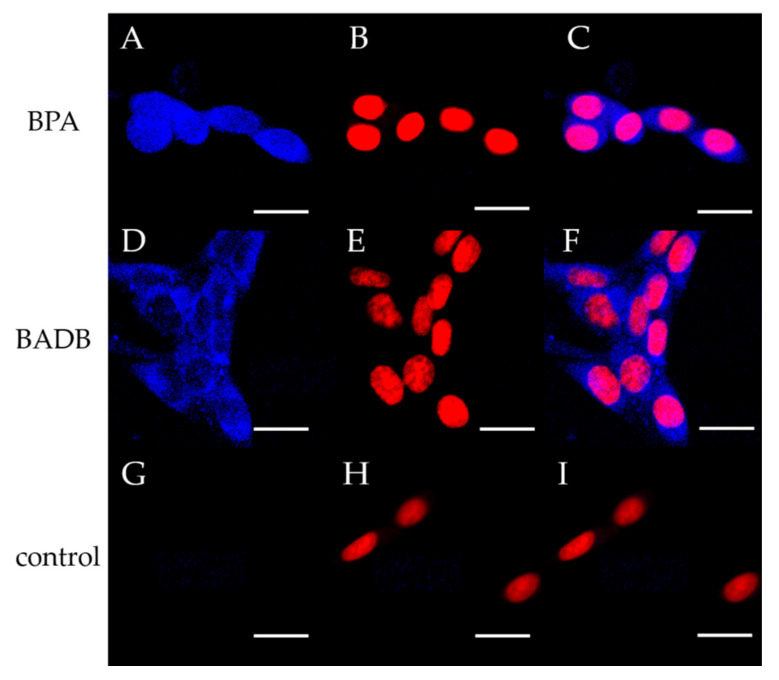
The micro-distribution of BPA and BADB in F98 cells. Scale bar = 20 μm (**A**): Fluorescence (ex: 405 nm) microscopy image showing the distribution of BPA over a 20-min period using DAHMI as the boron sensor. (**B**,**E**,**H**): Nucleus labeled with DRAQ5^TM^. (**C**) Merged image of (**A**,**B**). (**D**): Fluorescence (ex: 405 nm) microscopy image showing the distribution of BADB over a 20-min period using DAHMI as the boron sensor. (**F**): Merged image of (**D**,**E**). (**G**): Fluorescence (ex: 405 nm) microscopy image showing the distribution of medium without boron over a 20-min period using DAHMI as the boron sensor. (**I**) Merged image of (**G**,**H**).

**Figure 4 biology-09-00437-f004:**
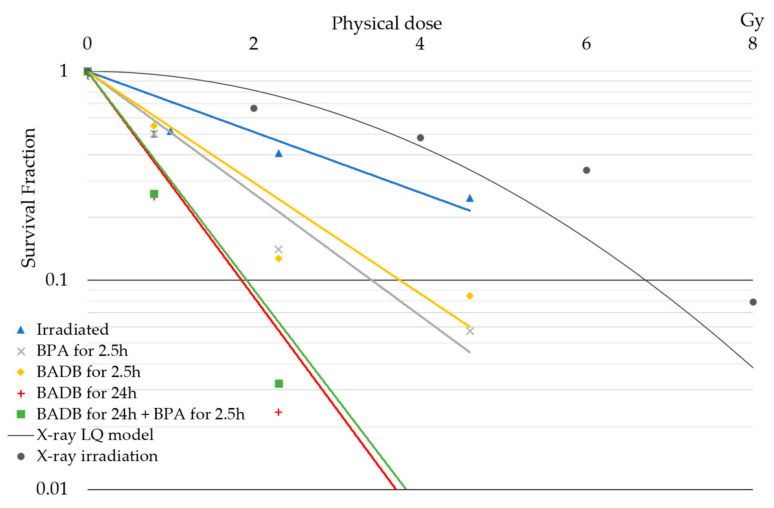
Clonogenic survival lines after neutron irradiation and photon irradiation fitted to the linear–quadratic (LQ) model for F98 glioma cells. Surviving fractions against physical doses were plotted to surviving approximate lines in each boron compound group. The error bars indicate standard deviation.

**Figure 5 biology-09-00437-f005:**
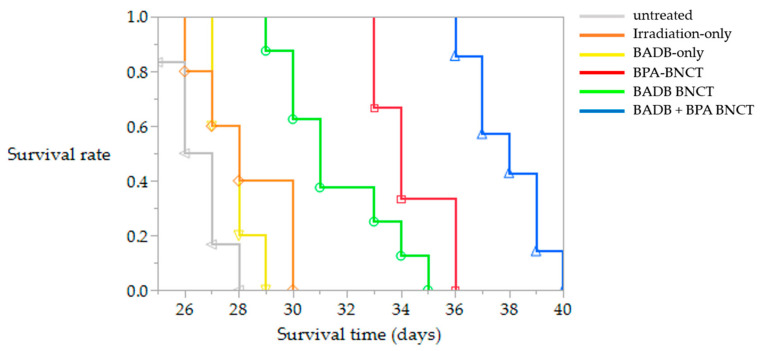
Kaplan–Meier survival curves of F98 glioma-bearing rats after neutron irradiation. Survival times in days after tumor implantation were plotted for the following groups: untreated controls (gray line), neutron irradiation-only (yellow line), BADB-only (orange line), BPA boron neutron capture therapy (BNCT) (red line), BADB BNCT (green line), and the combination of BADB and BPA BNCT (blue line).

**Table 1 biology-09-00437-t001:** Summary of biodistribution study with BPA and BADB in brain tumors and blood. i.v.: intravenous; CED: convection-enhanced delivery.

Boron Compound	Route	Dose ^a^ (mg B/kg)	Time ^b^ (h)	*n* ^c^	Boron Concentration ± SD (µg B/g)			Ratios ^d^	
					Tumor	Brain	Blood	T/Br	T/Bl
BADB (1000 µg B/mL)	CED (200 µL/24 h)	1.2	2	5	32.8 ± 16.7	1.0 ± 0.8	0.5 ± 0.2	32.8	65.6
			6	5	45.8 ± 16.5	1.3 ± 0.7	0.4 ± 0.1	35.2	114.5
			24	3	25.1 ± 26.6	0.2 ± 0.1	0.2 ± 0.0	125.5	125.5
BPA (1000 µg B/mL)	i.v.	12	2	4	17.8 ± 1.4	4.3 ± 0.9	8.7 ± 3.0	4.1	2.0
			6	4	13.4 ± 2.1	3.7 ± 0.6	6.1 ± 1.4	3.6	2.2

^a^ Dose in terms of the total amount of boron. ^b^ Time since the termination of boron compound (BADB or BPA) administration. ^c^ Number of animals. ^d^ T/Br indicates the tumor to normal brain (contralateral brain) ratio, T/Bl indicates the tumor to blood ratio.

**Table 2 biology-09-00437-t002:** The physical and photon-equivalent doses recorded during the neutron irradiation study. CBE: compound biological effectiveness.

Group	Physical Dose ^a^ (Gy)			Photon-Equivalent Dose ^d^ (Gy-Eq)			
	Brain	Tumor ^b^		Brain	Tumor ^b^		
		Mean ^c^			Mean ^c^		
		(Mean − SD)	(Mean + SD)		(Mean − SD)	(Mean + SD)	CBE
Irradiated	1.0	1.0		1.7	1.7		
BPA	1.8	3.4 (3.2−3.7)		2.9 *	6.7 (6.3−7.1)		2.02
BADB	1.4	2.5 (2.0−3.1)		(1.9) **	4.5 (3.5−5.5)		1.75
					9.1 (6.5−11.8)		4.66

^a^ Physical dose is a value calculated using the equation D_B_ + D_N_ + D_H_ + D_γ_. D_B_ is boron dose. D_N_ is nitrogen dose. D_H_ is the physical dose due to the elastic scattering between epithermal or fast neutrons and the hydrogen nucleus. D_γ_ is the measured dose of gamma rays mixed in the neutron beam. ^b^ The physical dose and photon-equivalent dose of BADB were based on a 25% boron contribution from CED administration. ^c^ The values of one SD lower and higher boron concentrations were calculated. ^d^ The photon-equivalent dose was a value calculated using the equation D_B_ × CBE + D_N_ × relative biological effectiveness of nitrogen (RBE_N_) + D_H_ × relative biological effectiveness of hydrogen (RBE_H_) + D_γ_ (RBE_N_ and RBE_H_: 3.0 [23]); photon-equivalent doses were calculated using the CBE based on the results of the colony-forming assay. * The CBE of 1.35 for the normal brain in BPA was used [24]. ** The CBE for the normal brain of BADB is unknown. We used the CBE for BPA as a reference.

**Table 3 biology-09-00437-t003:** Survival times of F98 glioma-bearing rats after neutron irradiation.

Group	*n* ^a^	Survival Time			%ILS ^b^
		Mean ± SD	Median	Range	Median
Control	6	26.5 ± 1.0	26.5	25−28	
Irradiated	5	28.2 ± 1.8	28	26−30	
BADB controls	5	27.8 ± 0.8	28	27−29	5.7%
i.v. BPA	6	34.3 ± 1.4	34	33−36	28.3%
BADB	8	31.6 ± 2.1	31 *	29−35	17.0%
BADB and i.v. BPA	7	38.0 ± 1.4	38 **	36−40	43.4%

^a^ Number of animals. ^b^ The percentage of increased lifespan (%ILS) was defined relative to the mean survival times of untreated controls. * i.v. BPA is significantly better than BADB in the median survival time (MST). (*p* < 0.05 by log-rank test). ** BABD and i.v. BPA is significantly better than i.v. BPA in the MST. (*p* < 0.05 by log-rank test).

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
