# Peer review of "The Therapeutic Effects of Dodecaborate Containing Boronophenylalanine for Boron Neutron Capture Therapy in a Rat Brain Tumor Model"

_biology, 2020, doi:10.3390/biology9120437_

Round 1
Reviewer 1 Report
This paper is very interesting, however it is not clear why authors used BADB administered by CED, probably due to blood-brain barrier (BBB) . Therefore, authors should show the result of bio-distribution study with BADB administered by i.v. in Table1.
Reviewer 2 Report
The article requires a complete revision by someone with good English knowledge, who understands BNCT and the experiments performed and can correctly reproduce what the authors want to express. It is not conducive to report all the many problems of this paper, because some of them are probably due to a misunderstandable translation. Because Prof. Barth was at some moment supporting this research activities, the authors might consider to ask him to overtake a co-authorship with the aim of improving the linguistic component and making the article easier to understand.
Here are some (non-exhaustive) comments on some aspects that should be taken into account in a revision:
- line 54: Never a (human) patient suffering for thyroid cancer has been treated by BNCT. The quotated publication by Pisarev and all does not report a clinical application
- the B-10 enrichment of the drugs is not reported. Example: BPA coerectly contains only one boron atom but not necessarily a B-10 atom (line 66)
- line 82: the quoted article (Hattori et al. [15]) does not exist
- The term "physical dose" is widely used in the slang of the BNCT community, but it is nonsense in scientific terms and should not be used in a scientific publication. Dose is always physical. A clear definition of the dose values used for the evaluation of the experiments is mandatory.
- Standard deviations should be added to fig. 4
- Table 3 "*p < 0.05 by log-rank test vs. the MST of i.v. BPA" is obviously not correct, the mean OST of the BPA group being higher as of the BADB group
Reviewer 3 Report
This paper showed the efficacy of new boron agents for BNCT.
Boronophenylalanine-amide alkyl dodecaborate (BADB) was tested in vitro and in vivo. BADB had additional effect on BPA-BNCT to prolong the survival periods.
Broad comments
The main text should be carefully edited by a native English speaker.
The reference should be carefully selected and checked, because I cannot find some reference from the website of the journals. i.e. ref [15]
The author should describe the structural formula of BADB, or clarify the reason why it cannot be described.
The purpose of this experiment should be added in material and methods.
The reviewer cannot understand how to decide the route and dose of DADB and BPA in each experiment.
Strength point was that, these experiments both in vivo and in vitro used real neutron source of BNCT and measured survival fraction or survival periods.
Weakness point was that, in each experiments, the control group settings are different, which is difficult to understand. Using CED.
Specific comments
P1 line 24
Simple summary will focus on your work. The first half of this chapter is redundant.
P1 line 43
What is the difference of 10B/mL and B/mL? DADB is 10B concentrated or not?
P2 line 82
I cannot find reference [15]. But the synthesis pathways and structural formulas of BADB are important for later discussion. It is recommended to specify them in the text.
In my understanding, BADB has boron cluster (12 boron) and also retained the original structure of the BPA containing a boron. MW 547.72 and boron content of 23.7% mean 13boron, isn’t it?
P2 line 87
No description of converting to a fructose complex.
P3 line 114
What was the rate of tumor engraftment? If it is not 100%, how to check the engraftment?
P3 line 136- P4 line 144
The authors use DAHMI as a boron detector, but this compound is not commercially available, No one can re-experiment and the detail of the settings are meaningless. It has been reported to react with the structure of boronophenylalanine. It is not clear that the reaction with the BADB is equivalent. Why the authors use different boron concentration of BPA and BADB?
P4 line 148
In this experiment, did the author try to reveal the additional effect of BPA?
Does the Group 5 use the additional BPA on BADB containing medium or remove BADB then use BPA incubation? Have you measured the boron concentration of the group 5?
P5 line228
Is the notation of the experimental result standard error or standard deviation?
Did the authors repeat the same experiment multiple times, or the distribution of multiple dishes spread from one irradiated tube, or the irradiation of multiple tubes?
Figure 1
Does Figure 1c without S.E or S.D means the experiment performed once?
P5 line 242 and Figure2
The difference between BPA and BADB was not recognize in the figure.
Add negative control. If it is not quantitative, it may be easier to understand by showing an expanded cell images.
P5 line 255
10ug10B/mL Boron exposure was different in setting of fugure 1. Why?
P6 line 276
The authors gave CED mediated tumor boron concentration. If main theme of this work is the efficacy of BADB, compare the same route of BPA (or Sodium borocaptate) mediated tumor boron concentration.
P7 line 299- Figure4
What was the cause of death? All of them were primary tumor death? Is there any oral / tracheal mucositis or sign of cachexia, especially in early death of BADB-BNCT group? How about Boron leakage to cerebrospinal fluid space within 24hrs?
P10 line 335
The description is too strong. How to solve the problem by BADB? Extracellular leakage was solved by CED in this article, and did not yet compare with BPA in same methods.
P10 line 336
Figure 2 does not add any information to the discussion.
P15 line 424
How to increase the penetration depth of thermal neutron?
Round 2
Reviewer 2 Report
There is an overall improvement in the language, but the discussion needs to be completely rewritten.
The points which I listed in my first review were only examples and not a complete list of all problems, because the main problem still is the poor command of the English language and therefore a discussion of the scientific content is not possible.The reader may suspect what is meant by a sentence like "BPA-BNCT is currently in the practical stage due to the conduct of conventional research and is expected to become a common radiation therapy option" but the English sentence itself is incomprehensible.
Another example: "The cellular uptake of BADB was comparable to that of boron since the exposure was based on the boron concentration." On one hand, uptake of "boron" never was investigated. On the other hand, the meaning of "since the exposure was based on the boron concentration" is unclear.
Such sentences are still in every paragraph and makes that the nice research you have performed si not understandable.
Finally, the overal conclusion "BADB is a novel boron compound that has a prolonged survival effect in BNCT treatment" is not supported by the results and also needs to be rephrased.
Reviewer 3 Report
Again, check the spell, grammar, and all of the references.
The authors were able to answer the editor's questions generally well.
Comments
If the authors say things from a clinician's point of view, the aim of this paper still have the problem of CED,
because this method has been applied in a few cases to humans who have pontine glioma or spinal cord tumors.
I heard that the application of BPA mediated BNCT is being applied for recurrent brain tumors in Japan. CED is not used as a standard therapy now, it is so challenging to apply to BNCT. It is highly doubtful how it can handle large doses and whether it can be placed after surgery for brain tumors.
I think you need more compelling results with a sophisticated design to license out to pharmaceutical companies.
